# Child Abuse and Neglect Awareness among Medical Students

**DOI:** 10.3390/children9060885

**Published:** 2022-06-14

**Authors:** Mohammad H. Al-Qahtani, Haitham H. Almanamin, Ahmed M. Alasiri, Mohammed H. Alqudaihi, Mohammed H. AlSaffar, Abdullah A. Yousef, Bassam H. Awary, Waleed H. Albuali

**Affiliations:** 1Department of Pediatrics, King Fahd Hospital of the University, College of Medicine, Imam Abdulrahman Bin Faisal University, Dammam 34211, Saudi Arabia; aaayousef@iau.edu.sa (A.A.Y.); bhawary@iau.edu.sa (B.H.A.); wbuali@iau.edu.sa (W.H.A.); 2College of Medicine, Imam Abdulrahman Bin Faisal University, Dammam 34211, Saudi Arabia; haithamhuss@me.com (H.H.A.); alasiria09@gmail.com (A.M.A.); mhd200091@gmail.com (M.H.A.); mhs-5005@hotmail.com (M.H.A.)

**Keywords:** child abuse, neglect, medical students, awareness

## Abstract

Physical, emotional, and sexual abuse and various forms of neglect of children have been encountered more frequently by healthcare providers, particularly physicians. However, mismanagement of child abuse and neglect (CAN) due to a lack of awareness of it can lead to substantial and serious consequences. This study primarily aimed to evaluate the awareness of CAN among medical students and compare it between preclinical and clinical males and females in Imam Abdulrahman Bin Faisal University in Dammam, Saudi Arabia. A cross-sectional study using a self-reported-based questionnaire was carried out to study child abuse and neglect awareness and compare preclinical and clinical male versus female medical students during the first semester in 2021/2022. The majority of the participants were aware of CAN (90.6%), agreed that CAN exists locally (96.6%), believed that CAN is important in the medical field (96.3%), and expressed the important role of physicians in participating in the management of CAN (84.3%). Some students did not know about the legislation of CAN in Saudi Arabia (15%). The results show a lack of exposure to real CAN cases (80.3%) and the need for more formal education (70.3%). In general, the students were comparable, but there were significant differences showing more awareness in female students compared to males and, similarly, more awareness in clinical-year students. Both clinical and preclinical medical students were aware of CAN, with some concerns regarding their competency in dealing with CAN. CAN should be given more weight in the medical school curriculum.

## 1. Introduction

Child abuse and neglect (CAN) is a major issue in children’s lives all over the world, affecting them in different ways and life aspects. CAN is defined as harm to children under the age of eighteen, categorized as: physical abuse, emotional abuse, sexual abuse, or neglect. Nonetheless, serious consequences of CAN, such as developmental, neurobiological, and financial problems, are overwhelming for children, their caregivers, and healthcare systems [1].

The prevalence of CAN globally is widespread, as the World Health Organization estimates that 3 out of 4 children are abused physically and emotionally. Moreover, adults with a history of sexual child abuse have been shown in the literature to be prevalent at 1:13 for men and 1:5 for women [2]. Child neglect is the most common form of child abuse in the United States and can coexist with other forms of abuse, accounting for approximately half of reported cases of CAN and 1500 fatalities annually [3]. In Saudi Arabia, a study with a 2043 sample size showed that psychological abuse is the most common type of abuse, accounting for 74.9%, followed by physical abuse at 57.5%, neglect at 50.2%, and sexual abuse at 14.0% [4].

Studies on awareness of CAN among physicians and medical students have shown that CAN cases are under-detected and under-reported and might be dealt with improperly. One study showed under-detection in up to 90% of affected cases, despite being very common [5]. Furthermore, a proportion of cases might be managed inappropriately by being under-reported by physicians due to, but not limited to, consideration of families’ aversion and lack of knowledge about referral procedures. According to another study, nearly half of physicians and students consider non-life-threatening abuse to be acts of discipline and culturally acceptable [6,7]. However, in Saudi Arabia, in a questionnaire-based study of 5075 employees working in public, private, and non-profit organizations, 79% of them stated that CAN exists in the Kingdom. On the other hand, only 42% of the participants knew that there are official agencies that protect children in Saudi Arabia [8].

Similar studies to ours tackled the same topic, including one on a comparison between interns and medical students in a local university in Saudi Arabia, which concluded that although medical students had adequate knowledge about CAN, they did not have good knowledge about reporting CAN cases [9].

Healthcare professionals, including doctors, should be prepared earlier in their medical schools and be provided early training to detect, document, and prevent all types of child abuse and neglect, especially the ones with worse and long-standing devastating effects on victims, such as sexual child abuse, which will enable the confident physician to train and prepare the surroundings of the expected victims [10].

In our study, we primarily aimed to evaluate the awareness of medical students, compare preclinical and clinical males versus females, determine their thoughts on the need for additional academic education in the college, and identify the most accessed source of information for medical students about CAN from their perspectives.

The main aim of this study was to evaluate awareness of child abuse and neglect among medical students and compare preclinical versus clinical and males versus females in the College of Medicine at Imam Abdulrahman Bin Faisal University in Dammam, Kingdom of Saudi Arabia, which is considered the first study of its kind at the level of the country.

## 2. Materials and Methods

A cross-sectional study using a self-reported-based questionnaire was carried out to study child abuse and neglect awareness and compare preclinical-year, defined as the 3rd college year, and clinical-year, defined as the 6th year and representing the final medical year, male and female medical students in the college of medicine in Imam Abdulrahman Bin Faisal University, Dammam, Kingdom of Saudi Arabia, during the first semester in 2021/2022.

A total of 505 medical students, both male and female in both the third (preclinical) and sixth (clinical) years, were invited to participate in this study.

Four male medical students from the sixth year were excluded due to being members of the research team.

### 2.1. Participants

The study included 381 participants with a mean (SD) age of 21.75 ± 1.6 years. A total of 202 (53.0%) of the participants were male, and 179 (47.0%) of them were females. The number of preclinical-year students was 191 (50.1%), and 190 (49.9%) of them were in the clinical (sixth) year. About 345 (90.6%) participants were aware of child abuse and neglect, and their most important source of information was social media (311; 37.6%), followed by their medical college education (187; 22.6%) (Table 1).

### 2.2. Instruments

The data were collected using a self-reported questionnaire, which was sent to the students on their mobile numbers, which were taken with permission from the college administration, and it was standardized and completed on Google Forms, an online computerized form.

### 2.3. Procedure

The medical students were contacted via their mobile numbers to participate in our study using a Google Forms link. This research was granted ethical approval from the institutional review board (IRB) for studies involving human participants in Imam Abdulrahman Bin Faisal University in Dammam, Saudi Arabia, under National Committee of BioEthics (NCBE) registration number HAP-05-D-003 and was given the IRB number IRB-UGS-2021-01-387.

### 2.4. Data Analysis

Data analysis was performed using the Statistical Package of Social Sciences (SPSS) version 26 (IBM Corp., Armonk, NY, USA). Normality was examined using the Shapiro–Wilk test. Continuous variables are summarized as means ± standard deviation (SD), while categorical variables are summarized using frequencies and percentages. A Chi-square test or Fisher’s exact test, whichever was appropriate, was used to test the association between variables. The level of significance was set at two-sided *p* < 0.05.

## 3. Results

A total of 381 students responded out of 505, accounting for a response rate of 75.45%. Appendix A shows the gender and academic year distributions of the students.

About 367 (96.3%) of the participants felt that child abuse and neglect is an important topic in the medical field. Only 75 (19.7%) of the participants had seen or encountered a case of child abuse and neglect during their medical studies. The number of participants who believed that child abuse and neglect exist in the Saudi population was 368 (96.6%). 

### 3.1. Participants’ Opinions on the Importance, Prevalence, and Legal Issues of Child Abuse and Neglect in Saudi Population with Their Expected Roles

The participants thought that child abuse and neglect are either common or very common in the Saudi population, with 174 (45.7%) and 64 (16.8%), respectively. Three hundred and seventy-two (97.6%) of the participants thought that child abuse and neglect would affect the child’s future (Table 2).

Most of the participants believed that, as future physicians, they would have a role in dealing with child abuse and neglect in any discipline. Fifty-seven percent of the participants were interested in dealing with child abuse and neglect. Only 110 (28.9%) of the participants had taken academic courses or sessions on child abuse and neglect in their medical college. Among those who had taken the academic courses or sessions, the majority of them felt that courses were sufficient, fair, good, or excellent, and 268 (70.3%) of all participants preferred to receive more courses or sessions on child abuse and neglect in the College of Medicine (Table 3).

There was a significant difference between males and females in their opinion as to whether the topic of child abuse and neglect is important, with 93.1% acceptance in males and 100% acceptance in females (*p* = 0.000). In terms of the participants’ opinions on whether child abuse and neglect exist in the Saudi population, there was a significant difference in the proportion of males (26; 12.9%) and females (38; 21.2%) who said it was very common, and the neutral category showed a great proportional difference, with 74 (36.6%) males and 41 (22.9%) females (*p* = 0.012). Most of the female participants (167; 93.3%) felt that, as a physician, they would have an important role in dealing with child abuse and neglect in all disciplines, with 154 (76.2%) males responding in this way (*p* = 0.000). There was a significant difference in the proportion of male (128; 63.4%) and female participants (140; 78.2%) who preferred to receive more courses and sessions about child abuse and neglect (*p* = 0.000).

Participants were asked their opinions on the legal systems that deal with child abuse and neglect and the preventive measures. Participants’ opinions on who deals with the issue of child abuse and neglect showed that most of them thought that social workers, physicians, or police deal with this problem, as indicated by 347 (25.5%), 329 (24.7%), and 306 (22.5%) of the participants, respectively.

The participants’ beliefs on whether there is a legal system—legislation—to deal with child abuse and neglect in our Kingdom showed a statistically significant difference between male and female participants, with 179 (88.6%) and 145 (81.0%), respectively (*p* = 0.038). The proportion of participants who agreed that children are mostly abused by their parents significantly differed between males (101; 50.0%) and females (118; 65.9) (*p* = 0.001). There was a significant difference in the proportion of male and female participants who thought child abusers have psychological issues.

Expected psychological disorders in abusers were thought to be common by 163 (42.8%) of the participants, followed by 119 (31.2%). Legislation and law enforcement were believed by 234 (61.4%) of the participants to be the most effective preventive measure for child abuse and neglect, followed by public education for 102 (26.8%) of the participants (Table 4).

The proportion of those with awareness of child abuse and neglect differed significantly between third-year (166; 86.9%) and sixth-year (179; 94.2%) participants (*p* = 0.015).

Most of the sixth-year participants (178; 93.7%) felt that, as future physicians, they would have an important role in dealing with child abuse and neglect in all disciplines, with 143 (74.9%) third-year participants expressing this opinion (*p* = 0.000). The proportion of participants who had taken a session or a course on child abuse and neglect differed significantly between the third-year (22; 11.5%) and sixth-year (88; 46.3%) participants (*p* = 0.000). The participants’ opinions on whether abusers have psychological disorders differed significantly for the option ‘always’ between third-year (45; 23.6%) and sixth-year (24; 12.6) participants, and for the option ‘sometimes’, the third-year (40; 20.9%) and sixth-year (79; 41.6%) participants differed significantly (*p* = 0.000). All of the results are represented graphically (Appendix A).

### 3.2. Opinions by Academic Year

For participants in the third academic year, there was a significant difference between males and females in their opinion as to whether the topic of child abuse and neglect is important, with 22 (87.9%) men and 92 (100%) women agreeing (*p* = 0.001). The participants’ opinions that child abuse and neglect exist in the Saudi population indicated that there was a significant difference in the proportion between men (10; 10.1%) and women (21; 22.8%) who said it was ‘very common’, and the ‘neutral’ category showed a great proportional difference, with 40 (40.4%) men and 20 (21.7%) women (*p* = 0.013). Most of the female participants (82; 89.1%) felt that, as a physician, they would have an important role in dealing with child abuse and neglect in all disciplines, with 61 (61.6%) men feeling this way (*p* = 0.000). There was a significant difference in the proportion between male (56; 56.6%) and female participants (73; 79.3%) who preferred to receive more courses and sessions about child abuse and neglect (*p* = 0.001). For participants’ opinions on who the abuser often is, the option ‘parents’ significantly differed between males (48; 48.5%) and females (58; 63.0%), and the option ‘relatives’ differed between males (24; 24.2%) and females (10; 10.9%) (*p* = 0.021). There was a significant difference in the proportion of male and female participants who thought that child abusers have psychological issues: the option ‘Commonly’ was selected by 51 (51.5%) of the males and 33 (35.9%) of the females, and the option ‘sometime’ was selected by 17 (17.2%) of the males and 23 (25.0%) of the females (*p* = 0.034).

Among sixth-year (clinical) students, there was a significant difference in the participants’ proportion of awareness of child abuse and neglect, with 93 (90.3%) males and 86 (98.9%) females (*p* = 0.012). Those participants who had taken a session or course on child abuse and neglect during their medical year showed a significant difference in their response to: How good was the course? The option ‘fair’ significantly differed between males (29; 53.7%) and females (16; 47.1%), and the option ‘good’ differed between males (7; 13.0%) and females (2; 5.9%) (*p* = 0.043). Male and female participants differed in their opinion that parents are potential abusers of children, with 53 (51.5%) males and 60 (69.0%) females (*p* = 0.028) (Table 5).

### 3.3. Opinions by Gender

The opinions of the male participants on whether the topic of child abuse and neglect is important in the medical field differed significantly between third-year (87; 87.9%) and sixth-year participants (101; 98.1%) (*p* = 0.008). The proportion of those who had encountered or seen a case of child abuse and neglect during their medical studies was low among the participants, but the proportion still significantly differed between third-year (12; 12.1%) and sixth-year (27; 26.2%) participants (*p* = 0.011). Most of the sixth-year participants (93; 90.3%) felt that, as a physician, they would have an important role in dealing with child abuse and neglect in all disciplines, with 61 (61.6%) third-year participants holding this opinion (*p* = 0.000). The proportion of participants who had taken a session or a course on child abuse and neglect differed significantly between third-year (10; 10.1%) and sixth-year (54; 52.4%) participants (*p* = 0.000). The proportion of students in the two study years who chose parents as the most likely potential child abuser was significantly different, with 48 (48.5%) in the third year and 53 (51.5%) in the sixth year, and the proportion of their second choice as ‘relatives’ differed, with 24 (24.2%) in the third year and 11 (10.7%) in the sixth year (*p* = 0.012). The participants’ opinions on whether abusers have psychological disorders differed significantly for the option ‘Commonly’ between third-year (51; 51.5%) and sixth-year (43; 41.7%) participants, and for the option ‘sometimes’, the third-year (17; 17.2%) and sixth-year (44; 42.7%) participants differed significantly (*p* = 0.001).

For females, the proportion of awareness of child abuse and neglect differed significantly between third-year (79; 85.9%) and sixth-year (86; 98.9%) participants (*p* = 0. 001). Most of the sixth-year participants (85; 97.7%) felt that, as a physician, they would have an important role in dealing with child abuse and neglect in all disciplines, with 82 (89.1%) third-year participants feeling this way (*p* = 0.044). The proportion of participants who had taken a session or a course on child abuse and neglect differed significantly between third-year (12; 13.0%) and sixth-year (34; 39.1%) participants (*p* = 0.000). The participants’ opinions on whether abusers have psychological disorders differed significantly for the option ‘always’ between third-year (29; 31.5%) and sixth-year (14; 16.1%) participants, and for the option ‘sometimes’, the third-year (23; 25.0%) and sixth-year (35; 40.2%) participants differed significantly (*p* = 0.031) (Table 6).

## 4. Discussion

Child abuse and neglect (CAN) is a common medical issue with significant consequences that are under-reported and under-detected, even by pediatricians [1,2,5,6,7,8] Awareness of CAN with several meanings has been defined as a core topic for dealing with this problem [5,6,7,8,11,12]. Our research showed that most of our participants were aware, according to our standards, but there are some concerning results. Some contributory factors to incompetency with CAN, such as a lack of knowledge about CAN legislation and its involvement, found in our study have been found in other studies too [9,13].

Healthcare provider competency in CAN issue skills is key for this important topic. A published Australian study in 2011 about CAN awareness illustrated that only 9.5% of participants considered themselves to be very well educated; however, in another study in Pakistan among 575 healthcare professionals, 450 (78.3%) agreed that CAN should be documented, while, on the contrary, 363 (58%) did not take any action in suspected cases [6]. In Saudi Arabia, variation was noted among pediatricians regarding CAN reporting, ranging from 43–82%, being more reported by those who had had a local education compared to those who received their education abroad [14]. Other studies, including one in Saudi Arabia, found similar findings that healthcare professionals might have poor knowledge about CAN reporting, which is a significant issue [9,15,16,17,18].

It is important to teach this topic to medical students, as they are future physicians, and poor knowledge and awareness of CAN in medical students will increase the number of CAN cases going unnoticed [7,9,16,19]. In the literature, one study in Saudi Arabia comparing CAN awareness among medical interns and students in a local university in 2021 stated that knowledge of reporting systems in medical students was inadequate, and in another one, although medical students took formal training on CAN, many of them agreed on needing further training in this field [9,17]. In Turkey, a study showed that due to cultural and traditional norms, medical students considered beating to be a form of disciplinary action, which tells us that awareness of forms of physical abuse is also important in dealing with CAN [7].

Most of our study’s students were decently aware of CAN, in which the majority believed that it exists in Saudi Arabia; additionally, most thought that it is highly prevalent in Saudi Arabia. Moreover, the majority agreed that CAN will affect the patient’s wellbeing in the future, which correlates with the insight on the importance of this topic. Consequently, most of the students believed that they would have a role in dealing with CAN in general. 

Despite the majority believing that they will have a role in dealing with child abuse and neglect in their future careers, only about half of the students of both genders were interested in dealing with such cases, which might reflect the students’ consideration and appreciation of how difficult and sensitive this topic will be as a highly demanding branch of medical practice. Since we did not include it in the questionnaire and the study objectives, we do not know if the hesitance of the students is because they do not want to or are not willing to interfere with the victim’s family, or maybe this is an effect of observation of practicing doctors, or it is due to socialization—acceptance in society of certain restrictive parental attitudes. Recently, Sara F. Owaidah et al. listed all of the possible causes affecting the reporting of suspected child maltreatment in Saudi Arabia [20].

Our study shows that the current academic sessions in the curriculum were not sufficient, as they were less frequently described as being high quality in our survey, along with a high demand to receive additional education. This finding was similar to that in other studies [11,13,14,15,16].

In addition to this, only about a quarter of the students had received knowledge about CAN from the medical college, as it was frequently described as fair or insufficient by them, illustrating the norm for the need for additional special sessions and courses about CAN in the college curriculum since it should be their most reliable source as future healthcare professionals. Several local and international studies found similar results [9,21,22,23,24,25,26]. Interestingly, one study published in 2003 gave junior physicians special training and education about CAN, in which almost two-thirds of them did not have any courses or sessions on it and found that the majority agreed that the additional education received met their needs and improved their ability to recognize and manage CAN cases [27]. On the practical side, only a few students from each group had encountered real cases during their medical studies, and although the clinical students encountered more real cases, there was no major difference in the number of cases encountered by each group between preclinical and clinical males and females. This finding indicates a significant lack of exposure to real cases, even for clinical students, which in turn makes the students, regardless of their academic year, less familiar with CAN and increases the knowledge gap, which should be at least compensated for by more clinically oriented hands-on courses in the medical school curriculum.

More education and exposure to CAN cases showed differences between clinical-year and preclinical-year medical students. Our questionnaire showed that more clinical-year students were aware of CAN and saw it as an important topic. The preclinical-year students encountered fewer cases of CAN than clinical-year students, as expected due to the exposure volume differences. Clinical-year students did have more sessions about CAN, which could be one of the reasons why more of them had insight into CAN’s importance compared to preclinical-year students. Both preclinical-year and clinical-year students agreed on who the abuser commonly is and whether the abuser is affected psychologically (this paragraph is added to this part to make it coherent with the same point of discussion).

Some results show a worrisome lack of knowledge, where a proportion of the students did not know there is a legal system in Saudi Arabia or who is involved in dealing with CAN, which could underlie incompetency in CAN reporting, as other studies found [6,8,9]. In addition to that, a minority of the participants thought that administrators have a role in dealing with CAN, while the majority saw legislation and law enforcement as the most effective preventive measure, which contradicts the fact that administrators have direct involvement in legal issues, which could possibly eventually lead to a lack of knowledge.

When it comes to the overall differences between males and females, females were more aware than males. The females showed a higher rate of awareness in all aspects, especially the importance of CAN, which all of them agreed on. On the contrary, a minority of males perceived it as unimportant. Moreover, females tended to perceive CAN issues to be more problematic, as they saw it as more prevalent and more associated with abusers’ psychological disorders. In addition, a greater preference for legislation and law enforcement as a preventive measure correlates with a higher awareness of females about the existence of CAN legislation in the country. These findings indicate that more focus should be assigned to males regarding education.

When comparing the students by their academic year, males and females in the preclinical year showed several differences; the first one was regarding the perception of the importance of CAN in the medical field, where agreement by females was 100% of the responses compared to 87.9% of males, which indicates less awareness of its importance in preclinical males. Regarding how common CAN is in the Saudi population, the thought that CAN is prevalent was greater among females at 68.5%, while 48.5% of the males held this belief, which is not different from the whole male versus female comparison. On the contrary, the idea of the presence of psychological disorders associated with abusers was recognized differently between sexes in preclinical years: 16.2% and 31.5% of males and females, respectively, believed that the association is always present. Furthermore, 51.5% of males and 35.9% of females thought that the association is common. These last two findings are different from the comparison of all males versus all females, in which the males perceived more of an association between abusers and psychological disorders, which could mean that there is a common concept among males. As future physicians, more females (89.1%) thought that they would have a role in dealing with CAN regardless of their future discipline, while fewer males (61.6%) thought similarly, which is a clear explanation for why fewer males see CAN as important. While only a minority of the preclinical students had taken courses or sessions on CAN, only a few of the females described courses as good or excellent. This not only means that all preclinical students need more education but also means that males seem to need more attention. This can precisely explain why we found a lower percentage of males who thought that CAN is important. With regard to the most effective preventive measure, legislation and law enforcement was the most chosen answer by both sexes, more by females (69.6%) compared to males (56.6%), showing more females had awareness regarding dealing with CAN cases, although fewer of them knew about the presence of legislation of CAN in Saudi Arabia. Thus, technically, males had better knowledge on this matter. However, some of both genders might eventually need to access such a system, which should be taught to them to ensure possible gaps in knowledge are filled.

In comparing clinical males versus females regarding awareness of CAN, more females (98.9%) were aware of CAN compared to males (90.3%). The majority of males (53.7%) thought that the college’s courses about CAN were fair, only a few of them (13.0%) thought that they were good, and none of them thought that they were excellent. On the other hand, 47.1% of females perceived them as fair, 5.9% perceived them as good, and 8.8% perceived them as excellent. These findings correlate with the lower awareness of males. In addition, more males (10.7%) knew about CAN legislation in Saudi Arabia versus 6.9% of females. However, males showed higher rates than females statistically in regard to CAN legislation, although both showed low percentages, which is insignificant in this comparison.

We did not specifically investigate gender-related causes of the different perspectives in terms of more females looking and thinking about this issue in comparison to males. Pinar Okur et al. published some of the gender differences regarding sexual abuse [28].

The differences between students of the same gender, either preclinical males versus clinical males or preclinical females versus clinical females, showed some statistically significant results. For males, more preclinical students described the association as more prevalent, while in females, it was the opposite, which may suggest how each sex recognizes this matter differently with clinical advancement. Regarding the importance of CAN, all females, both preclinical and clinical ones, agreed on it. In addition to that, for males, more clinical students (98.1%) agreed on its importance versus the preclinical students (87.9%), which is within our expectations due to the greater knowledge gained by clinical students. For both genders, the majority of clinical students agreed that they would have a role in dealing with CAN cases in the future, which expectedly correlates with their higher level of education. As for the education received, both genders of clinical students received more education and more value from their education in terms of its quality, which is within expectations.

The limitations in this study are due to the nature of the research question and the limited sample size. Thus, the results were proportionally limited to the sample size. Furthermore, this research was conducted using an online questionnaire, which may be of less quality compared to other questionnaire techniques. Nonetheless, the accessibility to all participants was limited due to the methodology being an online survey. Our study was implemented in only one medical college in Saudi Arabia, and it would be interesting to extend this study to all national medical colleges as a country-wide study.

## 5. Conclusions

Child abuse and neglect (CAN) is one of the most serious yet underestimated clinical issues with multidisciplinary involvement; it is not only specific and limited to pediatricians but can be encountered in any specialty that deals with the pediatric population.

The physician’s judgment in different specialties is the cornerstone in the integrated approach for the prevention, diagnosis, and management of such a sensitive, with potentially devastating consequences, problem. Medical students, as future physicians, should be prepared theoretically as well as clinically to deal confidently with such cases. The medical college curriculum should integrate this topic in well-prepared courses that engage medical students earlier in encountering the different clinical presentations of CAN and enable them to have the clinical as well as the managerial skills to prepare them for suspecting, detecting, and managing real-life cases in their different specialties.

Although our findings show good general awareness of CAN among medical students, they emphasize the need for more practical courses in the college curriculum and for having them engage in a practical manner to deal with different types of CAN cases.

Since this topic is important and relevant nationally and globally, we recommend carrying out this type of study in all or most of the medical colleges, nationally and internationally, to come up with much clearer needs and gaps that medical students might have to help college deans and faculty develop better clinical course designs in their medical colleges’ curricula.

## Figures and Tables

**Table 1 children-09-00885-t001:** Demographic Details of the Participants and Awareness on Child Abuse and Neglect.

Characteristic	N	%
Age (in years) Median		
(Interquartile range IQR)	22.0 (20.0, 23.0)	
Mean ± SD	21.75 ± 1.6	
Gender		
Male	202	53
Female	179	47
Academic year		
Third	191	50.1
Sixth	190	49.9
Are you aware about child abuse and neglect?		
No	36	9.4
Yes	345	90.6
Source of information on child abuse and neglect (Multiple response allowed)		
Social media	311	37.6
Medical college	187	22.6
Relatives and friends	178	21.5
Mass media	152	18.4

**Table 2 children-09-00885-t002:** Participant’s Opinion on the Importance and the Level of Prevalence of Child Abuse and Neglect in Saudi Population.

Characteristic	N	%
Do you think the topic on child abuse and neglect is important in medical field?		
No	5	1.3
Yes	367	96.3
Uncertain	9	2.4
Have you seen or encountered a case of child abuse and neglect during your medical study?		
No	306	80.3
Yes	75	19.7
Do you believe child abuse and neglect exists in Saudi’s population?		
No	6	1.6
Yes	368	96.6
Uncertain	7	1.8
How common do you think child abuse and neglect is in Saudi’s population?		
Rare	28	7.3
Not common (neutral)	115	30.2
Common	174	45.7
Very common	64	16.8
Do you think child abuse and neglect will affect his/her future?		
No	2	0.5
Yes	372	97.6
Uncertain	7	1.8

**Table 3 children-09-00885-t003:** Participant’s Opinion on Their Responsibility in the Issue and Importance of Courses or Sessions on Child Abuse and Neglect.

Characteristic	N	%
As a future physician, generally at any discipline, do you think you have a role with child abuse and neglect?		
No	11	2.9
Yes	321	84.3
Uncertain	49	12.9
Do you have an interest to deal with child abuse and neglect?		
No	63	16.5
Yes	217	57
Uncertain	101	26.5
Have you taken an academic course(s) or session(s) about child abuse and neglect in your medical college?		
No	271	71.1
Yes	110	28.9
How good do you think those academic course(s) or session(s) about child abuse and neglect were?		
Insufficient	19	17.3
Sufficient	20	18.2
Fair	56	50.9
Good	11	10
Excellent	4	3.6
Do you like to receive more courses and sessions about child abuse and neglect?		
No	33	8.7
Yes	268	70.3
Uncertain	80	21

**Table 4 children-09-00885-t004:** Participant’s Response on the Legal Ways Child Abuse and Neglect Is Dealt and Preventive Measures.

Characteristic	N	%
Who do you think deals with child abuse and neglect? (Multiple response allowed)		
Social workers	347	25.5
Physicians	329	24.2
Police	306	22.5
Nurses	209	15.4
Administrators	169	12.4
Do you think there is a legal system -legalization- to deal with child abuse and neglect in our kingdom?		
No	57	15.0
Yes	324	85.0
Who do you think abuse children the most (the abuser)?		
Parents	219	57.5
Relatives (excluding parents)	51	13.4
Housemaids	83	21.8
Drivers (e.g., school drivers or personal drivers)	12	3.1
Teacher	16	4.2
Do you think that the child abuser has a psychological disorder?		
Never	5	1.3
Rarely	11	2.9
Sometimes	119	31.2
Commonly	163	42.8
Always	69	18.1
Uncertain	14	3.7
What do you think is the most effective preventive measure for child abuse and neglect?		
Public education	102	26.8
Medical staff education	16	4.2
School education	29	7.6
Legalization and law enforcement	234	61.4

**Table 5 children-09-00885-t005:** Academic Year Response Comparison.

Characterestics	Gender		*p*-Value	Academic Year		*p*-Value
	Males	Females No:		3rd Year	6th Year	
	No: 202	No: 179		No: 191	No: 190	
	No. (%)	No. (%)		No. (%)	No. (%)	
Are you aware about child abuse and neglect?						
No	22 (10.9)	14 (7.8)	0.307	25 (13.1)	11 (5.8)	0.015
Yes	180 (89.1)	165 (92.2)		166 (86.9)	179 (94.2)	
Do you think the topic on child abuse and neglect is important in medical field?						
No	5 (2.5)	0 (0.0)	0.000 *	4 (2.1)	1 (0.5)	0.019 *
Yes	188 (93.1)	179 (100.0)		179 (93.7)	188 (98.9)	
Uncertain	9 (4.5)	0 (0.0)		8 (4.2)	1 (0.5)	
Have you seen or encountered a case of child abuse and neglect during your medical study?						
No	163 (80.7)	143 (79.9)	0.844	164 (85.9)	142 (74.7)	0.006
Yes	39 (19.3)	36 (20.1)		27 (14.1)	48 (25.3)	
Do you believe child abuse and neglect exists in Saudi’s population?						
No	6 (3.0)	0 (0.0)	0.060 *	4 (2.1)	2 (1.1)	0.386 *
Yes	192 (95.0)	176 (98.3)		182 (95.3)	186 (97.9)	
Uncertain	4 (2.0)	3 (1.7)		5 (2.6)	2 (1.1)	
How common do you think child abuse and neglect is in Saudi’s population?						
Rare	16 (7.9)	12 (6.7)	0.012	20 (10.5)	8 (4.2)	0.088
Not common (neutral)	74 (36.6)	41 (22.9)		60 (31.4)	55 (28.9)	
Common	86 (42.6)	88 (49.2)		80 (41.9)	94 (49.5)	
Very common	26 (12.9)	38 (21.2)		31 (16.2)	33 (17.4)	
Do you think child abuse and neglect will affect his/her future?						
No	2 (1.0)	0 (0.0)	0.631 *	1 (0.5)	1 (0.5)	0.122
Yes	196 (97.0)	176 (98.3)		184 (96.3)	188 (98.9)	
Uncertain	4 (2.0)	3 (1.7)		6 (3.1)	1 (0.5)	
As a future physician, generally at any discipline, do you think you have a role with child abuse and neglect?						
No	10 (5.0)	1 (0.6)	0	8 (4.2)	3 (1.6)	0.000
Yes	154 (76.2)	167 (93.3)		143 (74.9)	178 (93.7)	
Uncertain	38 (18.8)	11 (6.1)		40 (20.9)	9 (4.7)	
Do you have an interest to deal with child abuse and neglect?						
No	38 (18.8)	25 (14.0)	0.286	27 (14.1)	36 (18.9)	0.277
Yes	108 (53.5)	109 (60.9)		116 (60.7)	101 (53.2)	
Uncertain	56 (27.7)	45 (25.1)		48 (25.1)	53 (27.9)	
Have you taken an academic course(s) or session(s) about child abuse and neglect in your medical college?						
No	138 (68.3)	133 (74.3)	0.198	169 (88.5)	102 (53.7)	0.000
Yes	64 (31.7)	46 (25.7)		22 (11.5)	88 (46.3)	
How good do you think those academic course(s) or session(s) about child abuse and neglect were?						1.000 *
Insufficient	8 (12.5)	11 (23.9)	0.057 *	4 (18.2)	15 (17.0)	
Sufficient	14 (21.9)	6 (13.0)		4 (18.2)	16 (18.2)	
Fair	35 (54.7)	21 (45.7)		11 (50.0)	45 (51.1)	
Good	7 (10.9)	4 (8.7)		2 (9.1)	9 (10.2)	
Excellent	0 (0.0)	4 (8.7)		1 (4.5)	3 (3.4)	
Do you like to receive more courses and sessions about child abuse and neglect?						
No	28 (13.9)	5 (2.8)	0	17 (8.9)	16 (8.4)	0.438
Yes	128 (63.4)	140 (78.2)		129 (67.5)	139 (73.2)	
Uncertain	46 (22.8)	34 (19.0)		45 (23.6)	35 (18.4)	
Do you think there is a legal system -legalization- to deal with child abuse and neglect in our kingdom?						
No	23 (11.4)	34 (19.0)	0.038	35 (18.3)	22 (11.6)	0.065
Yes	179 (88.6)	145 (81.0)		156 (81.7)	168 (88.4)	
Who do you think abuse children the most (the abuser)?						
Parents	101 (50.0)	118 (65.9)	0.001	106 (55.5)	113 (59.5)	0.001
Relatives (excluding parents)	35 (17.3)	16 (8.9)		34 (17.8)	17 (8.9)	
Housemaids	43 (21.3)	40 (22.3)		35 (18.3)	48 (25.3)	
Drivers (e.g., school drivers or personal drivers)	9 (4.5)	3 (1.7)		10 (5.2)	2 (1.1)	
Teacher	14 (6.9)	2 (1.1)		6 (3.1)	10 (5.3)	
Do you think that the child abuser has a psychological disorder?						
Never	3 (1.5)	2 (1.1)	0.026 *	5 (2.6)	0 (0.0)	0.000 *
Rarely	7 (3.5)	4 (2.2)		8 (4.2)	3 (1.6)	
Sometimes	61 (30.2)	58 (32.4)		40 (20.9)	79 (41.6)	
Commonly	94 (46.5)	69 (38.5)		84 (44.0)	79 (41.6)	
Always	26 (12.9)	43 (24.0)		45 (23.6)	24 (12.6)	
Uncertain	11 (5.4	3 (1.7)		9 (4.7)	5 (2.6)	
What do you think is the most effective preventive measure for child abuse and neglect?						
Public education	63 (31.2)	39 (21.8)	0.047	50 (26.2)	52 (27.4)	0.919
Medical staff education	9 (4.5)	7 (3.9)		8 (4.2)	8 (4.2)	
School education	19 (9.4)	10 (5.6)		13 (6.8)	16 (8.4)	
Legalization and law enforcement	111 (55.0)	123 (68.7)		120 (62.8)	114 (60.0)	

*–Fisher’s exact test.

**Table 6 children-09-00885-t006:** Comparison of the Gender Response in Each Academic Year.

Characteristic	Academic Year		*p*-Value	Academic Year		*p*-Value
	3rd (N = 191)			6th (N = 190)		
	Male	Female		Male	Female	
	No. (%)	No. (%)		No. (%)	No. (%)	
Are you aware about child abuse and neglect?						
No	12 (12.1)	13 (14.1)	0.681	10 (9.7)	1 (0.5)	0.012
Yes	87 (87.9)	79 (85.9)		93 (90.3)	86 (98.9)	
Do you think the topic on child abuse and neglect is important in medical field?						
No	4 (4.0)	0 (0.0)	0.001 *	1 (1.0)	0 (0.0)	1.000 *
Yes	87 (87.9)	92 (100.0)		101 (98.1)	87 (100)	
Uncertain	8 (8.1)	0 (0.0)		1 (1.0)	0 (0.0)	
Have you seen or encountered a case of child abuse and neglect during your medical study?						
No	87 (87.9)	77 (83.7)	0.407	76 (73.8)	66 (75.9)	0.743
Yes	12 (12.1)	15 (16.3)		27 (26.2)	21 (24.1)	
Do you believe child abuse and neglect exists in Saudi’s population?						
No	4 (4.0)	0 (0.0)	0.152 *	2 (1.9)	0 (0.0)	0.251 *
Yes	93 (93.9)	89 (96.7)		99 (96.1)	87 (100)	
Uncertain	2 (2.0)	3 (3.3)		2 (1.9)	0 (0.0)	
How common do you think child abuse and neglect is in Saudi’s population?						
Rare	11 (11.1)	9 (9.8)	0.013	5 (4.9)	3 (3.4)	0.513 *
Not common (neutral)	40 (40.4)	20 (21.7)		34 (33.0)	21 (24.1)	
Common	38 (38.4)	42 (45.7)		48 (46.6)	46 (52.9)	
Very common	10 (10.1)	21 (22.8)		16 (15.5)	17 (19.5)	
Do you think child abuse and neglect will affect his/her future?						
No	1 (1.0)	0 (0.0)	0.684 *	1 (1.0)	0 (0.0)	0.707 *
Yes	94 (94.9)	90 (97.8)		102 (99.0)	86 (98.9)	
Uncertain	4 (4.0)	2 (2.2)		0 (0.0)	1 (1.1)	
As a future physician, generally at any discipline, do you think you have a role with child abuse and neglect?						
No	7 (7.1)	1 (1.1)	0.000 *	3 (2.9)	0 (0.0)	0.101 *
Yes	61 (61.6)	82 (89.1)		93 (90.3)	85 (97.7)	
Uncertain	31 (31.3)	9 (9.8)		7 (6.8)	2 (2.3)	
Do you have an interest to deal with child abuse and neglect?						
No	18 (18.2)	9 (9.8)	0.227	20 (19.4)	16 (18.4)	0.698
Yes	56 (56.6)	60 (65.2)		52 (50.5)	49 (56.3)	
Uncertain	25 (25.3)	23 (25.0)		31 (30.1)	22 (25.3)	
Have you taken an academic course(s) or session(s) about child abuse and neglect in your medical college?						
No	89 (89.9)	80 (87.0)	0.524	49 (47.6)	53 (60.9)	0.066
Yes	10 (10.1)	12 (13.0)		54 (52.4)	34 (39.1)	
How good do you think those academic course(s) or session(s) about child abuse and neglect were?						
Insufficient	2 (20.0)	2 (16.7)	0.775 *	6 (11.1)	9 (26.5)	0.043 *
Sufficient	2 (20.0)	2 (16.7)		12 (22.2)	4 (11.8)	
Fair	6 (60.0)	5 (41.7)		29 (53.7)	16 (47.1)	
Good	0 (0.0)	2 (16.7)		7 (13.0)	2 (5.9)	
Excellent	0 (0.0)	1 (8.3)		0 (0.0)	3 (8.8)	
Do you like to receive more courses and sessions about child abuse and neglect?						
No	15 (15.2)	2 (2.2)	0.001	13 (12.6)	3 (3.4)	0.076
Yes	56 (56.6)	73 (79.3)		72 (69.9)	67 (77.0)	
Uncertain	28 (28.3)	17 (18.5)		18 (17.5)	17(19.5)	
Do you think there is a legal system -legalization- to deal with child abuse and neglect in our kingdom?						
No	14 (14.1)	21 (22.8)	0.121	9 (8.7)	13 (14.9)	0.183
Yes	85 (85.9)	71 (77.2)		94 (91.3)	74 (85.1)	
Who do you think abuse children the most (the abuser)?						
Parents	48 (48.5)	58 (63.0)	0.021 *	53 (51.5)	60 (69.0)	0.028 *
Relatives (excluding parents)	24 (24.2)	10 (10.9)		11 (10.7)	6 (6.9)	
Housemaids	15 (15.2)	20 (21.7)		28 (27.2)	20 (23.0)	
Drivers (e.g., school or personal drivers)	7 (7.1)	3 (3.3)		2 (1.9)	0 (0.0)	
Teacher	5 (5.1)	1 (1.1)		9 (8.7)	1 (1.1)	
Do you think that the child abuser has a psychological disorder?						
Never	3 (3.0)	2 (2.2)	0.034 *	0 (0.0)	0 (0.0)	0.577 *
Rarely	5 (5.1)	3 (3.3)		2 (1.9)	1 (1.1)	
Sometimes	17 (17.2)	23 (25.0)		44 (42.7)	35 (40.2)	
Commonly	51 (51.5)	33 (35.9)		43 (41.7)	36 (41.4)	
Always	16 (16.2)	29 (31.5)		10 (9.7)	14 (16.1)	
Uncertain	7(7.1)	2 (2.2)		4 (3.9)	1 (1.1)	
What do you think is the most effective preventive measure for child abuse and neglect?						
Public education	27 (27.3)	23 (25.0)	0.097 *	36 (35.0)	16 (18.4)	0.057 *
Medical staff education	6 (6.1)	2 (2.2)		3 (2.9)	5 (5.7)	
School education	10 (10.1)	3 (3.3)		9 (8.7)	7 (8.0)	
Legalization and law enforcement	56 (56.6)	64 (69.6)		55 (53.4)	59 (67.8)	

*–Fisher’s exact test.

## Data Availability

Data are available on request from the corresponding author.

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
