# Peer review of "Child Abuse and Neglect Awareness among Medical Students"

_children, 2022, doi:10.3390/children9060885_

Round 1
Reviewer 1 Report
The article entitled “Child Abuse and Neglect Awareness Among Medical Students” presents a topic of great interest: The importance of knowledge of child abuse and maltreatment by medical students and future doctors.
In addition, the study carried out in an important country such as Saudi Arabia. Congratulations to the authors for becoming aware of such an important issue. If caregivers are not aware of child abuse this problem cannot change. It is critical that doctors have tools to detect abuse and neglect in childhood.
The statistical analysis and the study is simple but provides information of interest.
INTRODUCTION
The introduction is brief but appropriate.
I recommend the article:
Ortiz-Tallo M, Calvo I (2020) Child sexual abuse: Listening to the victims. Arch Community Med Public Health 6(2): 135-137.DOI: https://dx.doi.org/10.17352/2455-5479.000092
The article raises the need that Doctors and health professionals need to know this reality and be prepared to prevent and cope with abuse.
MATERIALS AND METHODS
I sugest that in point 2 of Materials and Method the authors organize it more clearly. For example:
2.1. Participants (here should be included the paragraph that appears after in Results “The study included 381 participants....”(Table 1). This table is not really required.
2.2. Instruments
2.3. Procedure (including ethical approval)
2.4. Data Analysis
RESULTS
It is presented too extensively by repeating all the data that are in the tables. As well as data that could go in other sections.
I recommend simplifying this section and improving the way of presenting it to make it more attractive.
The first sentence “The medical students were contacted via their mobile....” must be in Procedure.
The second sentence must go in the subsection Participants “The study included 381 participants....”(Table 1).
Table 1 completed must be commented on the “Participants” section.
I think it might be easier for the reader if you reorganized or titled the results from the beginnning. i.e.
1) Participant’s opinion on the importance and the level of prevalence of child abuse and neglect
2) Participant’s opinion on their responsibility in the issue
Analising if the comments in these sections can join two tables in some case.
it is not necessary to describe each table when presenting results in the text. The reader can see the results in the table and in the text the authors can refer that the results are in the table and highlight those they consider essential. This would make it easier to read.
CONCLUSION
Usually this last point is called Discussion.
In the discussion, the results are summarized, interpreted and extrapolated, their implications and limitations are analyzed, and the hypotheses raised are confronted, considering how the perspective of other authors has been.
It is recommended to rewrite this section.
Author Response
Thank you so much for your valid points and instructions
All the revised and edited parts of the manuscript are in RED
We did revised the manuscript so it fits the points you mentioned as follows;
- In INTRODUCTION we included the article you recommended to be integrated in the introduction and we added the reference as number 10
- In the METHODS we relabled the subtitles and did the appropriate changes.
- In RESULTS section we rewrite the text changes in the different paragraphes you mentioned
- We eliminate a lot of the explanatory test in the results section and we refer to the corresponding tables for the detailed informations.
- We combined table 5 and 6 in one new table labeled as table 5
- We eliminate table 8 and we were left with only 6 tables out of initial 8
- In CONCLUSION section we rewrote the paragraph and made it more informative and fits the conclusion idea and not as part of the discussion.
- Thank you so much again and hope you find the changes appropriate and matches your valuable concept.
Reviewer 2 Report
The goal indicated by the authors was achieved. The aim was to assess the awareness of child abuse and neglect that medical students have. A comparative analysis of the awareness of female students - and male students, and between third-year and sixth-year students was carried out.
The study used a simple tool - a questionnaire with answer options to choose from.
The reasoning is clear.
Interesting material was obtained for those responsible for educational programes at medical universities. Based on the results, it can be assumed that the students have relatively little experience with children who are victims of violence. The question arises whether few students recognize the manifestations of violence in child patients, or whether the students do not want to interfere with the family. Or maybe this is an effect ob observation of practicing doctors? Or maybe it is the effect of socialization - acceptance in society of certain restrictive parental attitudes? (Authors signal this. But it is effect of influence of culture.) It would be interesting to expand the research to include being a victim of childhood abuse by the student himself, that is, personal experiences. There is also a question about the prevalence of violence against children in Saudi Arabia - are the manifestations "balanced" in different environments / groups with different socio-economic status and about the relations between these groups.
It is worth emphasizing one more area of ​​application of the results, namely the intensification of the impact of sensitizing medical teams to violence against children and the development of procedures for dealing in such cases - or - if such procedures are - then their use. The authors refer to this issue in the text.
And the issue of undertaking social and educational influences shaping the greater sensitivity of adults to harm to the child (e.g. media, schools).
There were differences between men and women in the explored areas. No psychological tools were used and no variables that could potentially be important for the reported sex differences were studied. Such a trait may be, for example, empathy, emotional intelligence. Women showed greater sensitivity to signals of violence against children - the question is, is this an effect of a higher level of empathy? Or maybe own experiences of girls?
Unfortunately, no explanation for the gender differences in psychological factors has been sought. However, it is possible to try - on the basis of studies (psychological literature) on the differences between men and women - to formulate some explanations - and treat it as a starting point for further exploration.
However, authors are more interested in developing guidance for program developers, including introducing more workshops or specialized courses for students. This is undoubtedly an important point.
The text can be treated as a starting point for further research on the manifestations of child abuse and the possibility of doctors becoming more involved in limiting this negative phenomenon.
Author Response
Thank you so much for your valuable comments and instructions on our manuscript.
We did review the comments and did the proper changes to match your valid points and suggestions as follows;
- In the DISCUSSION part, your point is really valid and needs to be studied in a well-designed research to look for the possible reasons and we tried to elaborate more for the possible causes of why the students might hesitate to be involved practically in the deep issues of the child abuse and neglect and we added a reference ( 30) to explain the possible causes in Saudi Arabia society in regards of different areas, socioeconomic background, religious and social factors.
- In regard to your important research-question, which was not targeted in our paper; what could be different in the opinions of females versus males in dealing and thinking perspectives of a sensitive and challenging issue like child abuse and neglect, we listed some possible causes and we referred to an important article, reference (31) as it studied the different aspects of emotional social and thinking perspectives that might affect the gender-differences in disclosing such sensitive issue with massive psychological impact on the victims and their families.( By the way we kept all the terms in the text as MALES and FEMALES and removed the MEN and WOMEN from the text to keep the whole manuscript in a good coherence)
- We really thank you again for all the valid points and suggestions, hoping the revised manuscript will find your satisfaction and acceptance.
- Best regards